# Maternal mental health and nutritional status of infants aged under 6 months: A secondary analysis of a cross-sectional survey

**Mubarek Abera**[1,2]*, **Melkamu Berhane**[2,3], **Carlos S. Grijalva-Eternod**[4,5],
**Alemseged Abdissa**[2,6], **Nahom Abate**[7], **Endashaw Hailu**[7], **Hatty Barthorp**[8],
**Elizabeth Allen**[9], **Marie McGrath**[10], **Tsinuel Girma**[2,11], **Jonathan CK Wells**[12], **Marko Kerac**[4],
**Emma Beaumont**[13]

**1** Department of Psychiatry, Faculty of Medical Science, Institute of Health, Jimma University, Jimma, Ethiopia, **2** Jimma University Clinical and Nutrition Research Center (JUCAN), Jimma University, Jimma, Ethiopia, **3** Department of Pediatrics and Child Health, Faculty of Medical Science, Institute of Health, Jimma University, Jimma, Ethiopia, **4** Department of Population Health, London School of Hygiene and Tropical Medicine (LSHTM), London, United Kingdom, **5** UCL Institute for Global Health, London, United Kingdom, **6** Armauer Hansen Research Institute (AHRI), Addis Ababa, Ethiopia, **7** GOAL Ethiopia, Addis Ababa, Ethiopia, **8** GOAL Global, Carnegie House, Dublin, Ireland, **9** Department of Medical Statistics, Faculty of Epidemiology and Population Health, London School of Hygiene and Tropical Medicine (LSHTM), London, United Kingdom, **10** Emergency Nutrition Network, Oxford, United Kingdom, **11** Harvard Chan School of Public Health, Addis Ababa, Ethiopia, **12** Population, Policy and Practice Research and Teaching Department, UCL Great Ormond Street Institute of Child Health, London, United Kingdom, **13** Department of Infectious Disease Epidemiology and International Health, London School of Hygiene and Tropical Medicine (LSHTM), London, United Kingdom

* mubarek.abera@ju.edu.et, abmubarek@gmail.com

**Data Availability Statement:** We can provide data via the corresponding author or responsible body

## Abstract

Maternal/caregivers' mental health (MMH) and child nutrition are both poor in low- and middle-income countries. Links between the two are plausible but poorly researched. Our aim was to inform future malnutrition management programmes by better understanding associations between MMH and nutritional status of infants aged under six month (u6m). We conducted a health facility-based cross-sectional survey of 1060 infants in rural Ethiopia, between October 2020 and January 2021. We collected data on: MMH status (main exposure) measured using the Patient Health Questionnaire (PHQ-9) and infant anthropometry indicators (outcome); length for age Z-score (LAZ), weight for age Z-score (WAZ), weight for length Z-score (WLZ), mid upper arm circumference (MUAC), head circumference for age Z-score (HCAZ) and lower leg length (LLL). Analysis of secondary data using linear regression was employed to determine associations between the main exposure and outcome variables. The result showed infants' mean (SD) age was 13.4 (6.2) weeks. The median score for MMH problem was 0 (inter quartile range 0–2) points, and 29.5% and 11.2% reported minimal and mild to severe depression score of 1–4 and 5–27 points, respectively. Mean (SD) LAZ was -0.4 (1.4), WAZ -0.7 (1.3), WLZ -0.5 (1.2), MUAC 12.4 (1.3) cm, HCAZ 0.4 (1.3) and LLL 148 (13.9) mm. In adjusted linear regression analysis, minimal MMH problem was negatively associated with infant LAZ marginally (β = -0.2; 95% CI: -0.4, 0.00; p = 0.05) and LLL (β = -2.0; 95% CI: -3.8, -0.1; p = 0.04), but not with other anthropometric indicators. Statistically significant associations were not found between mild

on reasonable request but we cannot upload and share data online as we did not receive consent from the participants to share data online in this way. Contact address to the chair for Ethics Committee at Jimma University is: mio.ayana@ju. edu.et.

**Funding:** This work is funded and supported by the Eleanor Crook Foundation (ECF) (EPPHZR37) to MA, MB, CSG, AA, HB, EA, MM, TG, MK and EB. The funder had no role in study design, data collection and analysis, decision to publish, or preparation of the manuscript.

to severe depressive symptoms and infant anthropometric outcomes. In conclusion, only minimal, but not mild, moderate or severe, maternal/caregivers' depressive symptoms are associated with infant anthropometry outcomes in this data set. Whilst there is a plausible relationship between maternal mental health problems and offspring nutritional status, we did not observed this. Possible reasons include: PHQ-9 not suited to our population; and only a small number of participants reporting moderate to severe level of depression. Further research to investigate and understand the relationship and pathways between maternal mental health and offspring nutritional status is required.

## Introduction

Poor levels of maternal/caregiver's mental health (MMH) and poor child nutrition and growth are major public health problems in low-and middle-income countries (LMICs) [1, 2]. Poor MMH represents a decline in the status of mental well-being during pregnancy or within a year of childbirth [3]. Among various MMH problems, depression and anxiety during the pre- and post-natal periods are most common and increasing, particularly in LMICs. Systematic reviews and meta-analyses showed a prevalence of 25% [4] and 33% [5] peri- and post-natal depression, respectively in LMICs. Several factors such as pregnancy and child birth-related changes, role transition during and after pregnancy, miscarriage, still birth and child death or loss, intimate partner and domestic violence, low maternal social and economic capital, and poverty contribute to the high burden of MMH problems in LMICs [1, 6]. In most LMICs health care delivery systems, MMH problems are treated within the general mental health care system making the service inaccessible.

The burden of malnutrition and poor growth, manifested as stunting, wasting, and underweight, are highest in children aged under-five years in LMICs [2]. Though sometimes treated separately in prevention and treatment programmes, there are common features between these different manifestations of malnutrition, most notably a short-term high mortality risk [7] and long-term risks of poor developmental outcomes and non-communicable diseases [8, 9]. Stunting for example starts in utero and continues during childhood [10, 11]. Currently, nearly 150 million children aged under-five years are stunted, mostly in LMICs [12]. Stunting prevalence is as high as 50% in some settings [13], 41% in Sub Saharan Africa and 37% in Ethiopia in 2020 [2]. Wasting is another major challenge associated with high case fatality and contributing to some 800,000 child-deaths per year [14]. Risk factors for child malnutrition include infections, and poor social, economic and environmental factors [15].

Linkages between MMH and child nutritional status are plausible. Some evidence suggest that poor MMH is a risk factor for poor child nutrition and growth [16, 17] but such evidence, especially in infants aged under-6 months (u6m), is scarce and complex [18–21]. Some of these studies showed a negative impact of MMH problems on infant nutritional status while others reported a lack of association or presence of complex and indirect relationships between the two. More evidence on this is needed, especially since the new WHO guidelines on malnutrition highlight the need to consider MMH as part of care for infants u6m at risk of poor growth and development (as well as older children) [22]. There is increasing interest into how to integrate MMH and malnutrition programming [23] so evidence on this area are essential.

In this study, we hypothesise that poor MMH is associated with poor infant u6m nutritional status. Our aim was to examine associations between MMH problems and infant nutritional status to add new knowledge on existing evidence in a low income setting in order to inform and guide future malnutrition prevention and intervention programmes.

## Methods and materials

### Study design

This is a secondary analysis of a previously reported community-based health facility survey in Ethiopia [24]. The survey was carried out in order to inform planning for a cluster randomised controlled trial [25].

### Setting

The original survey included health facilities in Jimma zone (food secure) and Deder district (food insecure), Ethiopia. The total population of Jimma zone is estimated to be over 3.6 million while that of Deder district is approximately 360,980 people [26]. Both sites are predominantly agrarian, their livelihood is dependent on cash crop production such as coffee and khat followed by cattle rearing, and the two sites have similar ethnic and sociocultural backgrounds. In terms of health care coverage, Jimma zone has about 124 health centres and seven hospitals and Deder district has 8 health centres. The data was collected between 12 October 2020 and 29 January 2021.

### Participants

A consecutive sampling method, over a 2-week period, was employed in each of 18 health centres to recruit a total of 1060 infants aged u6m. Recruitment was from delivery, immunization, and growth monitoring services and under-five clinics. We included all participants available in the original dataset. The detailed procedures and main findings of the study on anthropometric outcomes has been previously reported [24].

### Measurements

**Main exposure variable.** MMH was assessed using the Patient Health Questionnaire (PHQ-9). Presence of mental health symptoms over the last 14 days on PHQ-9 were scored as symptom presenting 0 = not at all, 1 = over several days, 2 = more than half the days, and 3 = nearly every day. The raw sum score of individual PHQ-9 items ranges from 0 to 27 points. Risk of depression is classified as 0 (none), 1–4 (minimal), 5–9 (mild), 10–14 (moderate), 15–19 (moderately severe) and 20–27 (severe) depression [27]. These cut offs were used in the statistical analysis.

**Outcome variables.** Infant anthropometry/nutritional status were (length for age z-score (LAZ), weight for age z-score (WAZ), weight for length z-score (WLZ), mid-upper arm circumference (MUAC), head circumference for age z-score (HCAZ) and lower leg length (LLL)). Length was measured using the UNICEF length board to the nearest completed 0.1 cm. Weight was measured using a digital weight scale (Seca 354) to the nearest 5g if weight < 10kg or to the nearest 10g if weight ≥ 10kg. Measurement procedures followed the WHO Child Growth Standard protocols. All anthropometric indicators were measured in duplicate and averages were used in the analysis.

**Household related covariates.** Interviewer administered questionnaire was used to collect data on levels of maternal/caregiver education, household family size, numbers of dependent children aged <18 years, maternal/caregiver's age, grandmother family support, household wealth status, and water and sanitation hygiene (WASH) status.

**Infant related covariates.** Sex, age and date of birth, multiple birth (singleton, twin, triplet, etc), birth order, number of siblings and breastfeeding status were collected.

## Data analysis

The data was analysed using Stata (StataCorp. 2021) Statistical Software: Release 17.0 College Station, Texas 77845 USA: StataCorp LLC. Percentage and frequency were reported for categorical data while mean, standard deviation (SD), or median and inter quartile range (IQR) are presented for continuous data after checking normality of the distribution. Exclusive breastfeeding was considered if the mother/caregiver reported that the infant was entirely breastfed and was not given anything apart from breastmilk in the last 24 hour. Wealth index is computed by principal component analysis using ownership to house, diesel water pump, radio, television, watch/clock, mobile phone, non-mobile phone, refrigerator, table, chair, sponge matrix, straw matrix, grass matrix, electrical stove, kerosene stove, kerosene lamp, press lamp, motor bicycle, motor cycle, cart, car, bajaj (three wheel small vehicle), solar panel, mill house, electric power or generator, sweeping machine, coffee land, and khat land.

Anthropometric indicators were calculated as the average of a pair of measurements. Nutritional status for Z-scores were computed using length, weight, head circumference, age and sex of the infant based on the 2006 WHO Child Growth Standards, using the *zanthro* Stata command. Based on the WHO recommendation, LAZ<-6.0 or >6.0, WAZ <-6.0 or >5.0, WLZ<-5 or >5.0 and length <45cm were considered outlier and excluded from the analysis in the same order from the dataset.

To test for robust relationships between MMH and infant nutritional status, unadjusted and adjusted linear regression analysis were conducted by considering different sets of confounders (infant (age, sex, birth order, and breastfeeding status), maternal (age, educational status and presence of grandmother support) and household characteristics (family size, wealth index, and water, sanitation and hygiene status)). We reported 95% Confidence Interval (CI) and p-value to indicate presence and strength of association.

## Ethics

Ethical approval for the original data was granted by Jimma University Institutional Review Board (ref: IHRPGD/478//2020) and by the Research Ethics Committee at the London School of Hygiene and Tropical Medicine (ref 18022). Informed consent/permission was granted by the local Zonal health bureau. Written informed consent was obtained from the mother/caregivers (guardians) of every infant for participation by trained research nurses. This secondary analysis, looking at mental health in detail, is directly covered by the original protocol and ethics under the secondary objective "to assess the mental health status of mothers and caregivers of surveyed newborns and infants u6m."

## Results

### Participants' background

From the total of 1060 participants included in the original dataset, 1036 participants were included in the current analysis. Fig 1 shows the study participant flow chart.

Of those analysed, 577 (55.3%) infants were male and 262 (25.1%) were first born. Infant's mean age (SD) was 12.8 (6.2) weeks. Mean (SD) maternal/caregiver's age was 25.9 (5.4) years, and more than half of the mothers/caregivers 619 (59.4%) had attended at least some level of formal education. Nearly 60% of the households had a family size of five or less.

### Exposure and outcome characteristics

In terms of MMH, 920 (88.8%) of the mothers/caregivers scored none to minimal depression [614 (59.3% no depression and 306 (29.5%) minimal depression], while the rest 116 (11.2%)

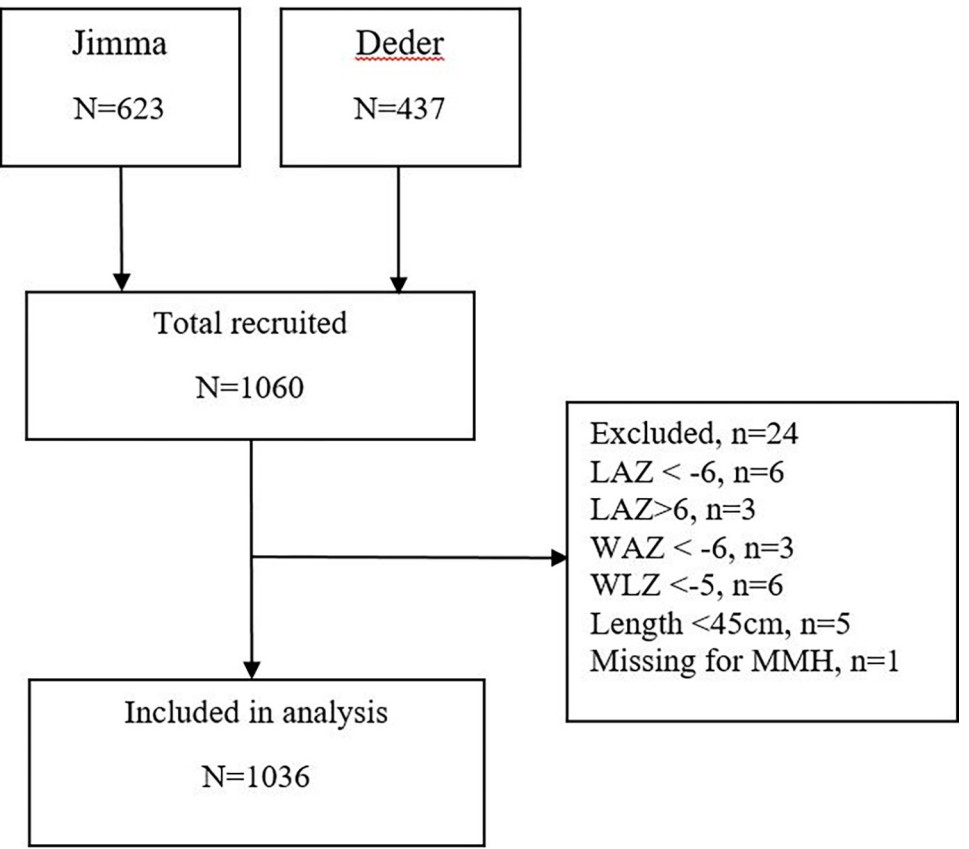

**Fig 1. Study participant flow chart.**

scored mild to severe depression [78 (7.5%) mild, 23 (2.2%) moderate, 8 (0.8%) moderately severe, and 7 (0.7%) severe depression]. Fig 2 shows the distribution of MMH outcome on the PHQ-9 score in a histogram. The median MMH score was 0 with an IQR of 0–2 points.

The overall mean and (95% CI) for infant nutritional status was LAZ -0.4 (-0.4, -0.3), WAZ -0.7 (-0.7, -0.6) and WLZ -0.5 (-0.6, -0.4). All were below the mean value for WHO reference data. The mean (95% CI) value for MUAC was 12.4 cm (12.3, 12.5), HCAZ 0.4 (0.3, 0.5) and LLL 148 mm (147.1, 148.9). Comparison for infants' nutritional status by level of maternal mental health is presented in Table 1. Fig 3 shows scatter plot between MMH score and infant LAZ.

## Association between MMH and infant nutritional status

In Table 2 we reported associations of MMH and infant anthropometric indicators. In unadjusted statistical analysis, only minimal depression (PHQ score 1–4) compared to no depression (PHQ score 0) was negatively associated with LAZ ($\beta$ = -0.2; 95% CI: -0.4, -0.01; P = 0.04) and LLL ($\beta$ = -2.0, 95% CI: -3.9, -0.1; P = 0.04), but not with other anthropometric indicators: WAZ ($\beta$ = -0.1; 95% CI: -0.3, 0.04; P = 0.12), WLZ ($\beta$ = 0.02; 95% CI: -0.2, 0.2; P = 0.83), MUAC ($\beta$ = -0.1, 95% CI: -0.3, 0.1; P = 0.37) and head circumference ($\beta$ = -0.1, 95% CI: -0.3, 0.1; P = 0.15). In the final adjusted statistical model, minimal maternal depression was associated negatively with LAZ ($\beta$ = -0.2; 95% CI: -0.4, 0.0; P = 0.05) marginally and LLL ($\beta$ = -2.0; 95% CI: -3.9, -0.1; P = 0.04), and PHQ score 1–27 was associated with LLL (($\beta$ = -1.8; 95% CI:

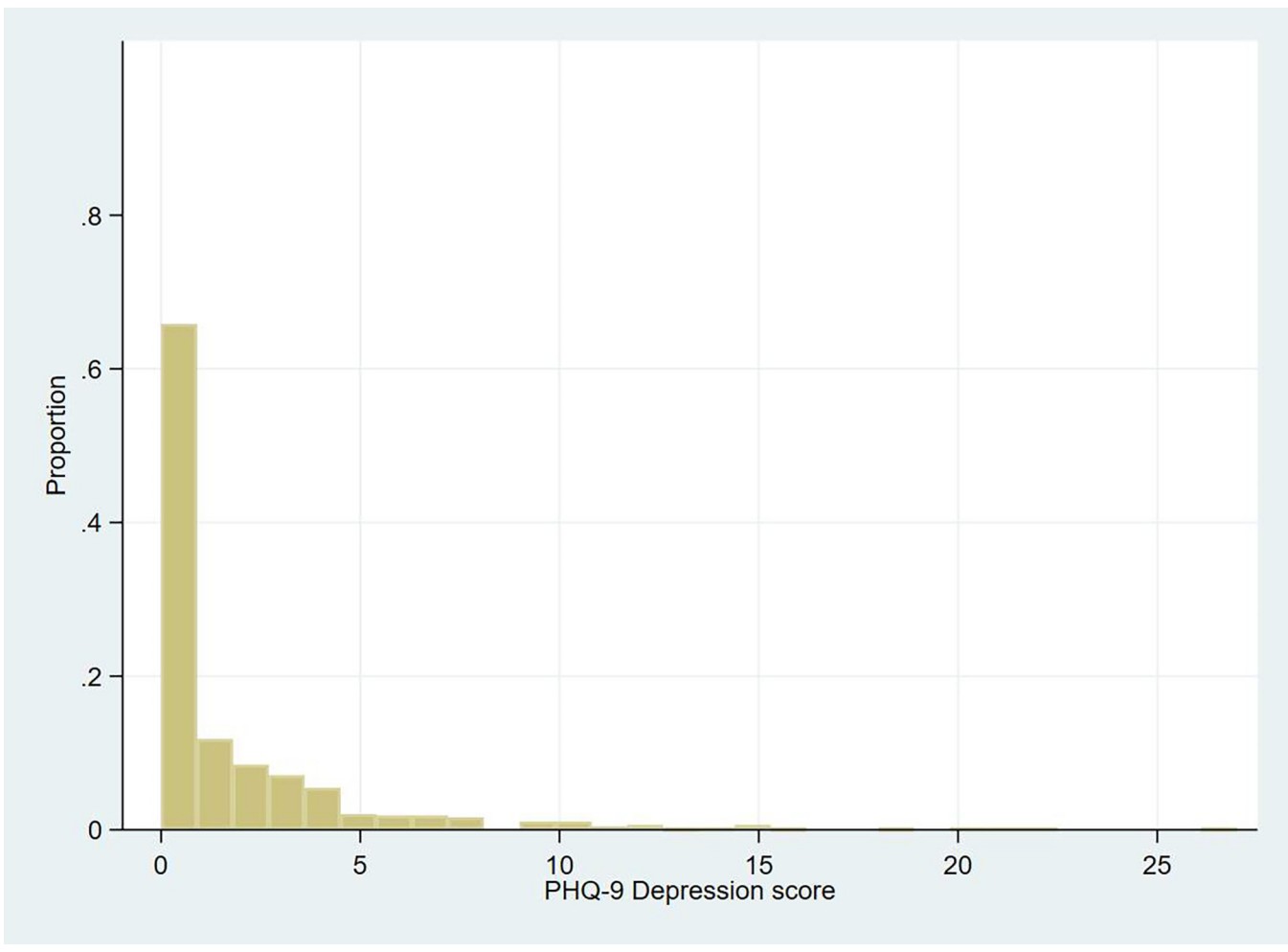

**Fig 2. Maternal mental health score histogram.**

-3.4, -0.1; P = 0.04). In both unadjusted and adjusted statistical models, significant associations were not found between mild, moderate, moderately severe, and severe level of depressions with any of infant anthropometric indicators (LAZ, WAZ, WLZ, MUAC, HCAZ and LLL).

Covariates positively associated with infant anthropometric indicators were higher wealth index with LAZ (β = 0.08, 95% CI: 0.03, 0.13; P = 0.003), WAZ (β = 0.12, 95% CI: 0.08, 0.17; P = 0.001), WLZ (β = 0.09, 95% CI: 0.05, 0.13; P = 0.001), MUAC (β = 0.06, 95% CI: 0.02, 0.11; P = 0.01), and HCAZ (β = 0.07, 95% CI: 0.03, 0.12; P = 0.002); increasing infant birth order with LAZ (β = 0.11, 95% CI: 0.03, 0.20; P = 0.01), WAZ (β = 0.10, 95% CI: 0.02, 0.17; P = 0.01) and HCAZ (β = 0.13, 95% CI: 0.05, 0.21; P = 0.001); higher maternal schooling with LAZ (β = 0.24, 95% CI: 0.05, 0.43; P = 0.01), WAZ (β = 0.24, 95% CI: 0.07, 0.41; P = 0.01) and MUAC (β = 0.19, 95% CI: 0.01, 0.36; P = 0.03); female sex with WAZ (β = 0.16, 95% CI: 0.01, 0.31; P = 0.04) and HCAZ (β = 0.16, 95% CI: 0.001, 0.31; P = 0.05); higher maternal age with LLL (β = 0.29, 95% CI: 0.07, 0.52; P = 0.011); and improved water, sanitation and hygiene status with MUAC (β = 0.07, 95% CI: 0.01, 0.12; P = 0.02) and LLL (β = 0.64, 95% CI: 0.04, 1.24; P = 0.04).

Covariates negatively associated with infant anthropometric indicators were female sex with MUAC (β = -0.33, 95% CI: -0.48, -0.18; P = 0.001) and LLL (β = -2.51, 95% CI: -4.15, -0.87; P = 0.003); higher household family size with WLZ (β = -0.08, 95% CI: -0.13, -0.02;

**Table 1. Background characteristics of study participants, n = 1036.**

| Characteristics | Category | n(%), mean ± SD | |
|---|---|---|---|
| Infant sex | Male | 574 (55.4) | |
| Infant age | Weeks | 13.4 ± 6.2 | |
| Infant birth order | 1st born | 259 (25.0) | |
| | 2nd born | 214 (20.7) | |
| | 3rd born | 156 (15.1) | |
| | 4th born | 127 (12.3) | |
| | 5th born | 116 (11.2) | |
| | >5th born | 164 (15.8) | |
| Primary caregiver's age, years | | 25.9 ± 5.4 | |
| Primary caregiver schooling | Ever attended school | 612 (59.1) | |
| Household size, family number | 1–4 | 441 (42.6) | |
| | 5–6 | 306 (29.5) | |
| | >6 | 289 (27.9) | |
| Maternal/caregivers mental health | PHQ score 0 | 614 (59.3) | |
| | PHQ score 1–4 | 306 (29.5) | |
| | PHQ score 5–9 | 78 (7.5) | |
| | PHQ score 10–14 | 23 (2.2) | |
| | PHQ score 15–19 | 8 (0.8) | |
| | PHQ score 20–27 | 7 (0.7) | |
| Infant nutritional status | All participants (N = 1036) | Poor maternal mental health (PHQ score 5–27) (n = 116) | Maternal mental health (PHQ score 0–4) (n = 920) |
| Weight | 5.62 ± 1.26 | 5.45 ± 1.49 | 5.64 ± 1.23 |
| Length | 59.65 ± 4.71 | 59.09 ± 5.68 | 59.72 ± 4.57 |
| Length for age Z-score | -0.35 ±1.4 | -0.25± 1.25 | -0.35± 1.37 |
| Weight for age Z-score | -0.65 ±1.3 | -0.65± 1.27 | -0.65± 1.27 |
| Weight for length for age Z-score | -0.48 ±1.2 | -0.59± 1.22 | -0.47± 1.24 |
| MUAC, centimetre | 12.4± 1.3 | 12.31± 1.34 | 12.45± 1.25 |
| Head circumference Z-score | 0.41± 1.3 | 0.52 ± 1.42 | 0.40± 1.25 |
| Lower leg length, millimetre | 148.07± 13.92 | 149.07± 16.54 | 147.06± 13.6 |

P = 0.01), MUAC (β = -0.06, 95% CI: -0.12, -0.001; P = 0.048) and LLL (β = -0.64, 95% CI: -1.29, 0.01; P = 0.05); exclusive breastfeeding with MUAC (β = -0.39, 95% CI: -0.55, -0.24; P = 0.001) and LLL (β = -7.37, 95% CI: -9.01, -5.75; P = 0.001); and grandmother family support with WAZ (β = -0.2, 95% CI: -0.3, -0.001; P = 0.05) and WLZ (β = -0.2, 95% CI: -0.4, -0.1; P = 0.01).

## Discussion

In this study we examined associations between maternal/caregiver's mental health problems with infant nutritional status as indicated by anthropometry indices. The prevalence of maternal depressive symptoms (PHQ score 5–27) is only 11.2% and the mean (SD) nutritional status of infants aged u6m for LAZ was -0.4 (1.4), WAZ -0.7 (1.3), WLZ -0.5 (1.2), MUAC 12.4 cm (1.3), HCAZ 0.4 (1.3) and LLL is 148 mm (14.0). The prevalence of maternal depressive symptoms in our data is lower than previous reports from Ethiopia and other LMIC studies [28, 29]. Mild, moderate, moderately severe and severe depression showed no association with any of the nutritional status indicators in this study. Only minimal depressive symptoms associated with low LAZ marginally and low LLL but not with other anthropometric indicators.

Y-axis: PHQ-9 score

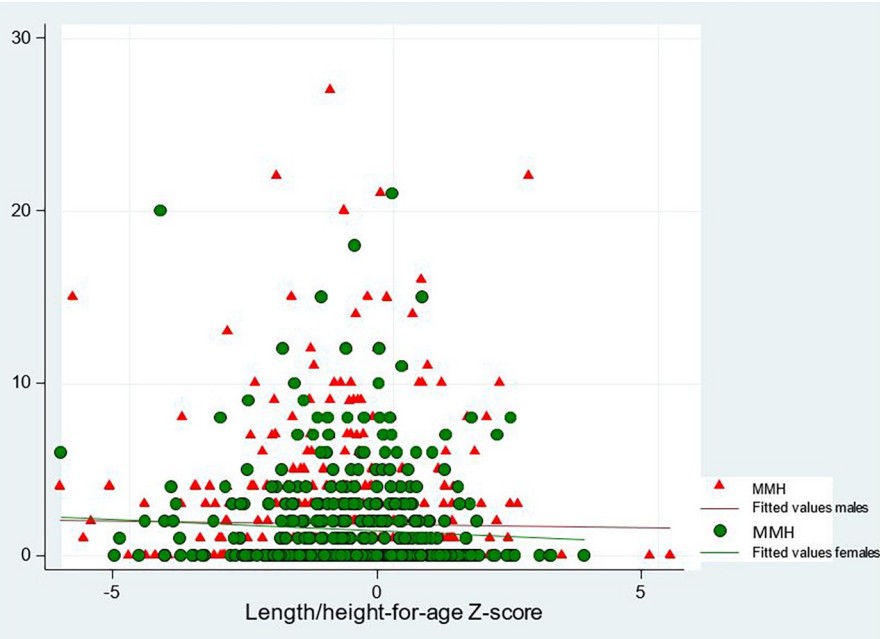

**Fig 3. Scatter plot between PHQ-9 score and infant length for age Z-score.**

The lack of associations between higher maternal depressive symptoms with infant nutritional status is consistent with previous studies where maternal common mental health problems were not associated with nutritional status of under five years children in Ethiopia [16, 30, 31], but contrasts with several other studies from LMICs [18–21, 32–37] including findings from Ethiopia [38, 39]. Our finding however is consistent with findings of high income settings (European cohort study [40] and the generation R study [41]) and upper middle income countries (e.g. Brazil) [42] where higher maternal depressive symptoms showed no association with children nutritional status [40–42]. In general, associations between postnatal MMH problems and poor child nutritional status have been mainly found in LMICs while lack of associations are mainly from high income settings [17]. Unlike our study, previous studies did not include tests of associations between minimal depressive symptoms and childhood nutritional status. Furthermore, the intensity of relationship and direction of associations between these two major public health problems are not well established.

Previously reported associations between maternal mental health problems and poor childhood nutritional status [18–21, 32–37] can be explained by several causal pathways involving poor maternal child care practice and episodes of childhood illness [5, 17, 43]. Maternal depressive symptoms have been associated with poor infant feeding practices [44], poor infant breastfeeding outcomes including shorter duration of exclusive breastfeeding [17], poor mother-to-infant bonding [17], low oxytocin total T4 concentration [45] and low breastmilk secretory immunoglobulin A concentration [46] in the breastmilk. Children especially in their early life are entirely dependent on their mother/caregiver's competencies and resources for nutrition, survival, growth and development. Maternal postnatal depression affects maternal ability and competences to provide responsive and supportive care to her baby [17]. These could result in poor maternal-child attachment, bonding, breastfeeding and stimulation; all of which could affect offspring nutrition and growth. Moreover, mothers with depression have

**Table 2. Associations between MMH and infant nutritional status in unadjusted and adjusted statistical models, n = 1036.**

| Unadjusted model | LAZ | | WAZ | | WLZ | | MUAC | | Head circumference | | Lower leg length | |
|---|---|---|---|---|---|---|---|---|---|---|---|---|
| | β | 95% CI (P-Value) | β | 95% CI (P-Value) | β | 95% CI (P-Value) | β | 95% CI (P-Value) | β | 95% CI (P-Value) | β | 95% CI (P-Value) |
| Unadjusted model -a | | | | | | | | | | | | |
| PHQ 1–4 | -0.20 | -0.38, -0.01 (0.04) | -0.14 | -0.31, 0.04 (0.12) | 0.02 | -0.15, 0.19 (0.83) | -0.08 | -0.25, 0.10 (0.37) | -0.13 | -0.31, 0.05 (0.15) | -1.95 | -3.86, -0.04 (0.04) |
| PHQ 5–9 | -0.05 | -0.36, 0.27 (0.78) | -0.03 | -0.33, 0.27 (0.85) | 0.005 | -0.29, 0.30 (0.97) | -0.15 | -0.44, 0.15 (0.34) | 0.08 | -0.22, 0.38 (0.60) | -1.92 | -5.20, 1.35 (0.25) |
| PHQ 10–14 | 0.35 | -0.22, 0.91 (0.23) | -0.05 | -0.58, 0.48 (0.85) | -0.49 | -1.00, 0.03 (0.06) | -0.15 | -0.68, 0.38 (0.58) | 0.11 | -0.42, 0.64 (0.69) | 2.12 | -3.68, 7.91 (0.47) |
| PHQ 15–19 | 0.09 | -0.87, 1.03 (0.86) | 0.30 | -0.59, 1.18 (0.51) | 0.28 | -0.59, 1.14 (0.53) | -0.23 | -1.11, 0.66 (0.62) | -0.02 | -0.91, 0.87 (0.97) | 1.91 | -7.80, 11.63 (0.70) |
| PHQ 20–27 | -0.17 | -1.19, 0.84 (0.74) | -0.27 | -1.22, 0.67 (0.57) | -0.14 | -1.07, 0.78 (0.76) | -0.38 | -1.32, 0.57 (0.44) | 0.01 | -0.94, 0.96 (0.98) | 3.07 | -7.30, 13.45 (0.56) |
| PHQ-1-27 | -0.13 | -0.30, 0.04 (0.12) | -0.11 | -0.26, 0.05 (0.18) | -0.01 | -0.16, 0.14 (0.91) | -0.10 | -0.26, 0.05 (0.20) | -0.07 | -0.23, 0.08 (0.36) | -1.56 | -3.29, 0.16 (0.08) |
| Unadjusted model -b | | | | | | | | | | | | |
| PHQ score 5–9 | 0.02 | -0.30, 0.33 (0.91) | 0.02 | -0.28, 0.31 (0.91) | -0.002 | -0.29, 0.28 (0.99) | -0.12 | -0.41, 0.17 (0.42) | 0.12 | -0.17, 0.42 (0.41) | -1.28 | -4.50, 1.95 (0.44) |
| PHQ score 10–14 | 0.41 | -0.15, 0.98 (0.15) | -0.01 | -0.53, 0.52 (0.99) | -0.49 | -1.01, 0.02 (0.06) | -0.12 | -0.65, 0.40 (0.65) | 0.15 | -0.38, 0.68 (0.58) | 2.76 | -3.01, 8.53 (0.35) |
| PHQ score 15–19 | 0.15 | -0.80, 1.10 (0.76) | 0.35 | -0.54, 1.23 (0.44) | 0.27 | -0.59, 1.13 (0.54) | -0.20 | -1.08, 0.68 (0.66) | 0.02 | -0.86, 0.91 (0.96) | 2.56 | -7.15, 12.27 (0.61) |
| PHQ score 20–27 | -0.11 | -1.12, 0.91 (0.84) | -0.23 | -1.17, 0.72 (0.64) | -0.15 | -1.07, 0.77 (0.75) | -0.35 | -1.29, 0.59 (0.47) | 0.05 | -0.89, 1.00 (0.91) | 3.72 | -6.65, 14.09 (0.48) |
| PHQ score 5–27 | 0.10 | -0.17, 0.36 (0.46) | 0.02 | -0.22, 0.27 (0.87) | -0.09 | -0.33, 0.15 (0.46) | -0.14 | -0.38, 0.10 (0.26) | 0.12 | -0.13, 0.36 (0.35) | 0.09 | -2.60, 2.78 (0.95) |
| Adjusted model–a | | | | | | | | | | | | |
| PHQ score 1–4 | -0.19 | -0.37, 0.001 (0.05) | -0.13 | -0.30, 0.04 (0.14) | 0.02 | -0.15, 0.19 (0.81) | -0.07 | -0.24, 0.09 (0.39) | -0.12 | -0.29, 0.05 (0.17) | -1.95 | -3.78, -0.13 (0.04) |
| PHQ score 5–9 | -0.04 | -0.36, 0.28 (0.80) | 0.03 | -0.26, 0.33 (0.84) | 0.09 | -0.20, 0.38 (0.56) | -0.11 | -0.40, 0.18 (0.48) | 0.08 | -0.21, 0.38 (0.58) | -2.34 | -5.48, 0.81 (0.15) |
| PHQ score 10–14 | 0.32 | -0.25, 0.88 (0.27) | -0.01 | -0.52, 0.51 (0.98) | -0.38 | -0.89, 0.13 (0.14) | -0.21 | -0.72, 0.31 (0.43) | 0.07 | -0.46, 0.59 (0.81) | 0.57 | -4.99, 6.13 (0.84) |
| PHQ score 15–19 | 0.02 | -0.93, 0.96 (0.97) | 0.28 | -0.59, 1.14 (0.53) | 0.33 | -0.52, 1.18 (0.45) | -0.16 | -1.02, 0.70 (0.71) | -0.09 | -0.97, 0.78 (0.84) | 2.71 | -6.56, 11.98 (0.57) |
| PHQ score 20–27 | -0.17 | -1.18, 0.84 ((0.74) | -0.14 | -1.07, 0.78 (0.76) | 0.04 | -0.87, 0.95 (0.93) | -0.42 | -1.33, 0.49 (0.37) | 0.003 | -0.94, 0.94 (1.00) | 0.80 | -9.10, 10.71 (0.87) |
| PHQ score 1–27 | -0.13 | -0.30, 0.04 (0.14( | -0.09 | -0.24, 0.07 (0.28) | 0.02 | -0.13, 0.17 (0.82) | -0.09 | -0.25, 0.06 (0.23) | -0.07 | -0.23, 0.09 (0.37) | -1.76 | -3.41, -0.11 (0.04) |
| Adjusted model -b | | | | | | | | | | | | |
| PHQ score 5–9 | 0.02 | -0.29, 0.33 (0.90) | 0.07 | -0.21, 0.36 (0.61) | 0.08 | -0.20, 0.36 (0.58) | -0.08 | -0.36, 0.20 (0.58) | 0.13 | -0.17, 0.42 (0.40) | -1.68 | -4.77, 1.41 (0.29) |
| PHQ score 10–14 | 0.38 | -0.18, 0.94 (0.19) | 0.04 | -0.48, 0.55 (0.89) | -0.39 | -0.90, 0.12 (0.13) | -0.18 | -0.69, 0.33 ((0.48) | 0.11 | -0.42, 0.63 (0.69) | 1.22 | -4.31, 6.75 (0.67) |
| PHQ score 15–19 | 0.08 | -0.86, 1.02 (0.87) | 0.32 | -0.54, 1.19 (0.47) | 0.32 | -0.53, 1.17 (0.46) | -0.13 | -1.00, 0.72 (0.76) | -0.05 | -0.93, 0.83 (0.91) | 3.38 | -5.88, 12.65 (0.47) |
| PHQ score 20–27 | -0.11 | -1.11, 0.90 (0.84) | -0.10 | -1.02, 0.82 (0.83) | 0.03 | -0.87, 0.94 (0.94) | -0.39 | -1.31, 0.52 ((0.40) | 0.04 | -0.89, 0.98 (0.93) | 1.47 | -8.43, 11.37 (0.77) |
| PHQ score 5–27 | 0.09 | -0.18, 0.35 (0.52) | 0.07 | -0.17, 0.32 (0.55) | 0.002 | -0.24, 0.24 (0.99) | -0.12 | -0.36, 0.12 (0.31) | 0.10 | -0.14, 0.35 (0.41) | -0.58 | -3.17, 2.01 (0.66) |
| Female infant | 0.16 | 0.01, 0.33 (0.58) | 0.16 | 0.01, 0.31 (0.04) | 0.10 | -0.05, 0.25 (0.20) | -0.33 | -0.48, -0.18 (0.001) | 0.16 | 0.001, 0.31 (0.05) | -2.51 | -4.15, -0.87 (0.003) |

*(Continued)*

**Table 2.** (Continued)

| Unadjusted model | LAZ | | WAZ | | WLZ | | MUAC | | Head circumference | | Lower leg length | |
|---|---|---|---|---|---|---|---|---|---|---|---|---|
| | β | 95% CI (P-Value) | β | 95% CI (P-Value) | β | 95% CI (P-Value) | β | 95% CI (P-Value) | β | 95% CI (P-Value) | β | 95% CI (P-Value) |
| Increasing birth order | 0.11 | 0.03, 0.20 (0.01) | 0.10 | 0.02, 0.17 (0.01) | 0.01 | -0.06, 0.09 (0.79) | 0.05 | -0.02, 0.12 (0.18) | 0.13 | 0.05, 0.21 (0.001) | 0.72 | -0.10, 1.54 (0.09) |
| Exclusively breastfed | 0.11 | -0.06, 0.28 (0.19) | 0.11 | -0.04, 0.26 (0.15) | 0.03 | -0.12, 0.18 (0.68 | -0.39 | -0.56, -0.24 (0.001) | 0.04 | -0.11, 0.20 (0.57) | -7.37 | -9.01, -5.75 (0.001) |
| Increasing primary caregiver/mother's age | -0.02 | -0.04, 0.01 (0.19) | -0.01 | -0.03, 0.01 (0.52) | 0.01 | -0.01, 0.03 (0.34) | 0.02 | -0.01, 0.04 (0.14) | -0.006 | -0.03, 0.02 (0.58) | 0.29 | 0.07, 0.52 ((0.011) |
| Increasing primary/ mother's schooling | 0.24 | 0.05, 0.43 (0.01) | 0.24 | 0.07, 0.41 (0.01) | 0.07 | -0.10, 0.24 (0.44) | 0.19 | 0.01, 0.36 (0.03) | 0.12 | -0.06, 0.30 (0.18) | 1.04 | -0.83, 2.90 (0.28) |
| Increasing household family size | -0.001 | -0.07, 0.07 (0.98) | -0.05 | -0.11, 0.01 (0.11) | -0.08 | -0.13, -0.02 (0.01) | -0.06 | -0.12, -0.001 (0.048) | -0.05 | -0.11, 0.01 (0.12) | -0.64 | -1.29, 0.01 (0.05) |
| Grand parent support available | -0.01 | -0.19, 0.17 (0.89) | -0.16 | -0.33, -0.001 (0.05) | -0.22 | -0.38, -0.06 (0.01) | -0.10 | -0.26, 0.06 (0.23) | -0.09 | -0.26, 0.08 (0.29) | -0.55 | -2.30, 1.21 (0.54) |
| Increasing wealth status | 0.08 | 0.02, 0.13 (0.003) | 0.12 | 0.08, 0.17 (0.001) | 0.09 | 0.05, 0.14 (0.001) | 0.06 | 0.02, 0.11 (0.01) | 0.07 | 0.03, 0.12 (0.002) | 0.10 | -0.58, 0.39 (0.70) |
| Increasing WASH status | 0.004 | -0.06, 0.06 (0.89) | 0.01 | -0.04, 0.07 (0.70) | 0.01 | -0.05, 0.06 (0.76) | 0.07 | 0.01, 0.12 (0.02) | 0.01 | -0.05, 0.06 (0.85) | 0.64 | 0.04, 1.24 (0.04) |

Note: model–a (comparator no depression) and model- b (comparator no to minimal depression); adjusted models are accounted for infant sex, birth order, exclusive breastfeeding status, mother/caregivers' age and schooling, household family size, availability of grand parent support, wealth status, and WASH status. Outputs for covariates are from final model-b.

poor emotional sensitivity and responsiveness to their environment including their babies [17] which may put at risk infant without support. Nurturing care as proposed by UNICEF, World Bank and WHO for child growth and development requires 5 domains of care (good health, adequate nutrition, safety and security, responsive care and early opportunity for learning) [47]. All of these domains require good maternal mental health and competences to manage holistic aspects of her child's condition.

However these causal pathways through which MMH problems impact offspring nutritional status and our proposed hypothesis are not supported in the current findings. Moreover the lack of association between MMH problems and poor child nutritional status in the previous three studies from Ethiopia [16, 30, 31] and from upper middle income and high income countries were [40–42] not adequately discussed. Several factors such as the setting and context how the data was collected, types of screening tools and strategies used, and the skills, competences and confidence of data collectors in asking sensitive questions in mental health to elicit symptoms could have all played a role for the low prevalence of depression and lack of association between MMH problems and poor child nutritional status in our data.

## Implication of the findings

Despite plausible relation between MMH problems and poor infant/child nutritional status exist, our data do not support this. The lack of association by itself in the current study however is not evidence of an absent link between the two problems. The low prevalence of moderate to severe depression in the current study could have lowered the statistical power to detect significant association. Depressive disorder by its nature is unique in its clinical presentation compared to many other psychiatric illnesses in that most patients with depression have a good level of insight and understanding about their poor mental health status. This may lead to under or over reporting based on the context and circumstances how, by whom and where

the screening process takes place. The high prevalence of minimal depressive symptoms over the low prevalence of moderate to severe depression in this study may indicate that the majority of severely affected participants could minimize symptoms due to lack of confidence to report symptoms in a non-conducive crowded screening environment. Screening for depression requires an established rapport and therapeutic relationship which could take a longer time or repeated visits to encourage clients to report their problems, trust the healthcare worker and overcome perceived or actual stigma and discrimination. These could have implications for future integration of MMH care into the maternal and child health care which are often be crowded and lack space to ensure adequate assessment, privacy and confidentiality. While there is increasing interest into how to integrate MMH and malnutrition programming [23], screening strategies for MMH in the management of small and nutritionally at-risk infants aged under 6 months and their mothers (MAMI) may require a particular adapted approach rather than simply applying a routine mental health screening approach to be acceptable, feasible and effective in routine care settings. In general additional qualitative, epidemiological and experimental studies are essential to understand the various circumstances that could explain the current findings to reach on a robust conclusion.

## Strengths and limitations

Strengths include the focus on young infants u6m where there is a lack of evidence in this area to support a clear decision making and action orientated process. We covered two different geographic settings representing food secure and insecure areas. Moreover we included a large number of study participants. This study however is not without limitations. The study is a facility based survey, conducted in specific season of the year during October–January and therefore the findings may not represent the general population in all seasons of the year. Potential confounders and effect modifiers such as maternal HIV/AIDS status, nutritional status, obesity as well as infant birth weight, gestational age and infant morbidities were not included in the study. Moreover the setting where MMH data were collected within the maternal and child health unit may not be conducive enough to establish a trusted relationship between participants and study nurses. Finally, despite the PHQ-9 has been widely used in LMIC settings, including in Ethiopia, it may not be optimal in identifying MMH symptoms in the MAMI context and other screening tools or strategies may be more valid and may show a different association with anthropometric deficits.

## Conclusions

Mild to moderate and severe levels of maternal/caregiver's mental health problems were not associated with poor infant nutritional status in this data set. Whilst there is plausible relationship between MMH problems and offspring nutritional status, this hypothesis was not supported in the current study. Thus, further research to investigate and understand the relationship and pathways between postnatal maternal mental health problems and infant u6m nutritional status is required both in Ethiopia and other LMIC settings.

## Supporting information

**S1 Checklist. Inclusivity in global research.**
(DOCX)

## Author Contributions

**Conceptualization:** Mubarek Abera, Melkamu Berhane, Carlos S. Grijalva-Eternod, Alemseged Abdissa, Hatty Barthorp, Elizabeth Allen, Marie McGrath, Tsinuel Girma, Jonathan CK Wells, Marko Kerac, Emma Beaumont.

**Data curation:** Mubarek Abera, Melkamu Berhane, Carlos S. Grijalva-Eternod, Alemseged Abdissa, Nahom Abate, Endashaw Hailu, Hatty Barthorp, Marie McGrath, Tsinuel Girma, Marko Kerac, Emma Beaumont.

**Formal analysis:** Mubarek Abera.

**Funding acquisition:** Mubarek Abera, Melkamu Berhane, Alemseged Abdissa, Hatty Barthorp, Marie McGrath, Tsinuel Girma, Jonathan CK Wells, Marko Kerac.

**Investigation:** Mubarek Abera, Carlos S. Grijalva-Eternod, Marie McGrath, Marko Kerac, Emma Beaumont.

**Methodology:** Mubarek Abera, Melkamu Berhane, Carlos S. Grijalva-Eternod, Alemseged Abdissa, Hatty Barthorp, Elizabeth Allen, Marie McGrath, Tsinuel Girma, Jonathan CK Wells, Marko Kerac, Emma Beaumont.

**Project administration:** Mubarek Abera, Melkamu Berhane, Carlos S. Grijalva-Eternod, Nahom Abate, Endashaw Hailu, Hatty Barthorp, Marie McGrath, Marko Kerac.

**Resources:** Mubarek Abera, Melkamu Berhane, Carlos S. Grijalva-Eternod, Hatty Barthorp, Marko Kerac.

**Software:** Mubarek Abera.

**Supervision:** Mubarek Abera, Melkamu Berhane, Carlos S. Grijalva-Eternod, Nahom Abate, Endashaw Hailu, Hatty Barthorp, Marko Kerac, Emma Beaumont.

**Validation:** Mubarek Abera, Marko Kerac, Emma Beaumont.

**Visualization:** Mubarek Abera, Marko Kerac, Emma Beaumont.

**Writing – original draft:** Mubarek Abera.

**Writing – review & editing:** Mubarek Abera, Melkamu Berhane, Carlos S. Grijalva-Eternod, Alemseged Abdissa, Nahom Abate, Endashaw Hailu, Hatty Barthorp, Elizabeth Allen, Marie McGrath, Tsinuel Girma, Jonathan CK Wells, Marko Kerac, Emma Beaumont.

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
