## [Decision Letter · Decision Letter 0]

7 May 2024

PGPH-D-24-00706

Maternal mental health and nutritional status of infants aged under 6 months: a secondary analysis of a cross-sectional survey

Dear Dr. Mubarek Abera,

Thank you for submitting your manuscript to PLOS Global Public Health. After careful consideration, we feel that it has merit but does not fully meet PLOS Global Public Health’s publication criteria as it currently stands. Therefore, we invite you to submit a revised version of the manuscript that addresses the points raised during the review process.

We look forward to receiving your revised manuscript.

Kind regards,

Abu Sayeed, MSc

Academic Editor

Journal Requirements:

2. Please amend your detailed online Financial Disclosure statement. This is published with the article. It must therefore be completed in full sentences and contain the exact wording you wish to be published.

a) State the initials, alongside each funding source, of each author to receive each grant. For example: "This work was supported by the National Institutes of Health (####### to AM; ###### to CJ) and the National Science Foundation (###### to AM)."

3. Please update your online Competing Interests statement. If you have no competing interests to declare, please state: “The authors have declared that no competing interests exist.”

4. Please provide separate figure files in .tif or .eps format only and ensure that all files are under our size limit of 10MB.

5. Please include a separate legend or caption for each figure in your manuscript.

Additional Editor Comments (if provided):

Reviewers' comments:

Reviewer's Responses to Questions

**Comments to the Author**

1. Does this manuscript meet PLOS Global Public Health’s publication criteria? Is the manuscript technically sound, and do the data support the conclusions? The manuscript must describe methodologically and ethically rigorous research with conclusions that are appropriately drawn based on the data presented.

Reviewer #1: Yes

Reviewer #2: Partly

Reviewer #3: Yes

2. Has the statistical analysis been performed appropriately and rigorously?

Reviewer #1: Yes

Reviewer #2: No

Reviewer #3: Yes

3. Have the authors made all data underlying the findings in their manuscript fully available (please refer to the Data Availability Statement at the start of the manuscript PDF file)?

Reviewer #1: No

Reviewer #2: No

Reviewer #3: No

4. Is the manuscript presented in an intelligible fashion and written in standard English?

Reviewer #1: Yes

Reviewer #2: No

Reviewer #3: Yes

5. Review Comments to the Author

Reviewer #1: The authors present a cross-sectional study investigating the relationship between maternal mental health, as determined using the PHQ-9 instrument, and infant nutritional status by anthropometry. Contrary to similar studies in low- and middle-income countries (LMIC), their analyses showed no association. The paper is clear and well-written. I have only minor comments.

The exposure, outcome and co-variables are on the whole well-defined and the methodology and results clearly reported. The results are discussed in the context of Ethiopian, and to a lesser extent, other LMIC reports.

Abstract:

‘u6m’ should be spelled out on first use.

Introduction:

The definition of poor MMH (line 2-3) should be referenced.

Methods:

A list of recruitments sites is presented. Please clarify how these sites were sampled/weighted as each is likely to represent a different group of infants. For example, neonates at delivery sites, well-children at immunization sites, malnourished children attending growth-monitoring services etc. What were the inclusion and exclusion criteria?

Wealth index is noted in the Abstract and Tables but it is not defined.

Please indicate how the data on household and other covariables were collected (self/interviewer-administered questionnaire?)

Certain maternal and infant characteristics that have been shown to affect infant growth have not been included. E.g., maternal comorbidities, including HIV status; maternal nutritional status, including obesity; infant birth weight and gestational age. These are important co-variables and if the data are not available, this lack should be included as a limitation.

Results:

In Table 1, the results of the PHQ-9 scores are presented as means and SD. In Figure 1, the distribution is clearly non-parametric and medians and IQR would be more appropriate. This applies to the distribution of all the continuous variables.

Other than this, I am not certain that the Figures offer additional benefit.

Implications of FINDINGS (typo)

Limitations:

Please include the lack of relevant potential confounders and effect modifiers as indicated above.

References:

Please include url and Access Date for on-line references.

Reviewer #2: I have annotated the pdf of the manuscript with my comments.

Main comments:

1. General comment: English needs a thorough check as the English is not of a good standard, with many grammatical and some spelling mistakes. I have corrected some as I went along but I would recommend that the manuscript is given to a scientific editor to read through and improve the English.

2. The anthropometrical measurements are meaningless given that the population is "under 6 months" and these measurements vary considerably during this period, hence I would remove them, e.g. lower leg length. The z-scores are acceptable as they correct for age (and sex).

3. The results in the text and table for the statistical analysis are incomplete and lack any mention of statistical significance. An association without a mention of a P-value is not helpful for interpretation and can be misleading to the reader. These need to be included as, without them, the data underlying the findings is incomplete.

Reviewer #3: The authors present a secondary analysis investigating the associations between maternal mental health and infant growth outcomes. This is an important topic and an. Area of much interest. They use a relatively large cohort with good characterization of maternal and infant characteristics/outcomes. The rate of mothers with depression was surprisingly low within the cohort which may limit some of the applicability of findings.

Please see specific comments below:

-Comment 1: It is unclear from the abstract that this was a secondary analysis. This should be clarified so readers understand the limitations.

-Comment 2: The abstract is very dense and contains too many results making it difficult for the reader to get through. I would suggest editing the abstract to be more succinct and present the main findings of the manuscript, not merely describing every finding.

-Comment 3: Setting section- Are there other important differences in the two districts that may influence the results? Are the populations similar ethnically and culturally or are there distinct practices that may make them distinct?

-Comment 4: Discussion- As the authors note, the prevalence of maternal depressive symptoms is much lower than many other published studies. Do the authors have a hypothesis as to why this is the case in the cohort? The PHQ-9 has been validated for use in many settings including Ethiopia; however, this may not translate to every region/ ethnic group. Can the authors speak to the validity of the PHQ-9 among the specific population that they screened? Can we be reassured that the results are valid, especially given the somewhat surprising low prevalence?

-Comment 5: The authors note that the lack of associations with maternal depression and infant nutritional outcomes; however, do the authors believe that they have adequate power in the current analysis given the lower than expected rate of maternal depression?

-Comment 6: The data was collected only between October and January and therefore do not reflect the full seasons/year. MMH may vary seasonally because of external factors, food security, etc. Do the authors believe that this is not a issue and the data is representative?

-Comment 7: In the conclusion, I think that it is important to state that "...[MMH] problems are not associate..." in this data set. The lack of findings here does not mean that there is no association. The conclusion should clearly state this.

-Comment 8: Please clarify in the footnote of Table 2 what the bolded values represent. This is not currently clear. Additionally there are some 95% CI values that are bolded without bolding of the beta-coefficient. Ensure that this was intentional.

6. PLOS authors have the option to publish the peer review history of their article (what does this mean?). If published, this will include your full peer review and any attached files.

**Do you want your identity to be public for this peer review?** For information about this choice, including consent withdrawal, please see our Privacy Policy.

Reviewer #1: No

Reviewer #2: No

Reviewer #3: No

---

## [Editor Report · Decision Letter 1]

29 Jul 2024

PGPH-D-24-00706R1

Maternal mental health and nutritional status of infants aged under 6 months: a secondary analysis of a cross-sectional survey

Dear Dr. Mubarek Abera,

Thank you for submitting your manuscript to PLOS Global Public Health. After careful consideration, we feel that it has merit but does not fully meet PLOS Global Public Health’s publication criteria as it currently stands. Therefore, we invite you to submit a revised version of the manuscript that addresses the points raised during the review process.

We look forward to receiving your revised manuscript.

Kind regards,

Abu Sayeed, MSc

Academic Editor
---

## [Editor Report · Decision Letter 2]

20 Aug 2024

Maternal mental health and nutritional status of infants aged under 6 months: a secondary analysis of a cross-sectional survey

PGPH-D-24-00706R2

Dear %Mubarek Abera%,

We are pleased to inform you that your manuscript 'Maternal mental health and nutritional status of infants aged under 6 months: a secondary analysis of a cross-sectional survey' has been provisionally accepted for publication in PLOS Global Public Health.

Best regards,

Abu Sayeed, MSc

Academic Editor